# OpenReview forum: "Language-Agnostic Bias Detection in Language Models with Bias Probing"
_EMNLP/2023/Conference — EMNLP 2023 Findings_

### Official Review · Reviewer_W1HN · 2023-07-24

**Soundness:** 4

**Excitement:**

4: Strong: This paper deepens the understanding of some phenomenon or lowers the barriers to an existing research direction.

**Paper Topic And Main Contributions:**

The authors propose LABDet, a language-agnostic method of measuring social bias in pretrained language models. They point out that prior methods (namely, evaluating perplexity over minimal pairs) are sensitive to minor perturbations in input. They mitigate this by training a sentiment classifier by adding a layer to the language model, and then computing the difference in resulting sentiment scores for neutral-sentiment minimal pairs. They show their results on a dataset for nationality, analyze the correlation with sentiment in the language model's training data, and measure variability across languages to understand the robustness of their method.

**Reasons To Accept:**

This paper makes progress in measuring social bias in language models that is less vulnerable to small perturbations in input. They also point out the importance of correlating the results with bias in the training dataset and collaborating with political scientists to do that.

**Reasons To Reject:**

As the authors mention, there are limitations involving the templates used. Moreover, they also point out that there may be forms of bias that are not captured by this method. It would be good to mention somewhere that even if this method yields the result that it cannot detect bias in the model, that does not mean that the model is unbiased.

**Reproducibility:**

5: Could easily reproduce the results.

**Reviewer Confidence:**

4: Quite sure. I tried to check the important points carefully. It's unlikely, though conceivable, that I missed something that should affect my ratings.

---

> ### Author Rebuttal · Authors · 2023-08-28
>
> We thank the reviewer for the valuable comments and will update our paper
> accordingly.

---

### Official Review · Reviewer_TUSA · 2023-08-05

**Soundness:** 3

**Excitement:**

3: Ambivalent: It has merits (e.g., it reports state-of-the-art results, the idea is nice), but there are key weaknesses (e.g., it describes incremental work), and it can significantly benefit from another round of revision. However, I won't object to accepting it if my co-reviewers champion it.

**Paper Topic And Main Contributions:**

This paper proposes a language-agnostic bias detection method for nationality in monolingual PLMs in Arabic, Dutch, English, French, German, and Turkish. First, they use templates containing positive/negative adjectives without any nationality information to train a sentiment classifier on top of PLMs. Then they use templates and corpus examples with a nationality slot, e.g., This [Nationality] person is neutral, to obtain the sentiment score for different nationalities. By subtracting absolute sentiment scores from the sentiment score of a neutral example without nationality, e.g., This [MASK] person is neutral, the relative sentiment score indicates the effect of nationality on the sentiment.

**Questions For The Authors:**

A. In Table 2, I am curious about why use different corpus templates for various languages instead of employing the same templates as shown in Table 1.

**Reasons To Accept:**

1. The paper is clear and well-structured.
2. It is interesting to explore the bias towards minority groups or nationalities associated with conflict and tension across different languages and show some interesting findings, like "large minority groups who produce content in the target language tend to have more positive sentiment, such as Turks in German PLMs and Senegalese/Moroccans in French PLMs".

**Reasons To Reject:**

1. Limited task and dataset. The experiments are limited to only one task - sentiment classification - and one dataset - SST2. To gain deeper insights, it would be better to explore the effects on different tasks or utilize multiple datasets for the same task.
2. Lack of implementation details. For example, the details of training classifiers of the multilingual models are missing.

**Reproducibility:**

3: Could reproduce the results with some difficulty. The settings of parameters are underspecified or subjectively determined; the training/evaluation data are not widely available.

**Reviewer Confidence:**

3: Pretty sure, but there's a chance I missed something. Although I have a good feel for this area in general, I did not carefully check the paper's details, e.g., the math, experimental design, or novelty.

---

> ### Author Rebuttal · Authors · 2023-08-28
>
> We thank the reviewer for the valuable comments.
>
> > “In Table 2, I am curious about why use different corpus templates for various
> languages instead of employing the same templates as shown in Table 1.”
>
> Corpus examples in Table 2 are extracted from their monolingual Wikipedia corpus
> by querying nationality terms. Therefore, we cannot select the same templates
> for various languages.
>
> For all experiments, we only employ templates from Table 1. We only compare
> corpus examples to templates in "5.3. Robustness Evaluation - Template vs.
> Corpus Examples" to show our method’s robustness to the use of minimal pair
> templates other than those that we designed ourselves.
>
> > “Lack of implementation details”
>
> We use the same implementation setup for all languages, described in L289-L296
> for training, L297-L314 for sentiment surfacing/inference with the specific PLMs
> for each language (L329-L333). We will release the reproducible codebase in the
> camera ready version.
>
> > “The experiments are limited to only [...] one dataset - SST2.”
>
> We use SST2 for the pretraining data analysis in 5.1 while we mostly use our
> templates for sentiment training in the rest of the experiments.

---

### Official Review · Reviewer_qjcz · 2023-08-05

**Soundness:** 4

**Excitement:**

3: Ambivalent: It has merits (e.g., it reports state-of-the-art results, the idea is nice), but there are key weaknesses (e.g., it describes incremental work), and it can significantly benefit from another round of revision. However, I won't object to accepting it if my co-reviewers champion it.

**Paper Topic And Main Contributions:**

The paper proposes a method to detect the nationality bias present in a pre-trained Language Model. To this end, the LM is fine-tuned for sentiment analysis on a corpus that is designed not to have any nationality bias (by manually creating the samples from templates) and then tested on neutral sentences containing nationality information. The method uncovers biases in LMs that are consistent with expectations from political science.

**Questions For The Authors:**

a) Since the models are only trained on positive and negative samples (no neutral ones), does this training induce a bias in itself? I don't consider this a problem, since there seem to be clear tendencies of the model to associate a nationality with either positive or negative sentiment, but it will likely still push the model away from the (correct) neutral predictions. Or is this even intentional, to force the model to show its bias?

**Reasons To Accept:**

a) Simple, but effective method

b) Generalisable to any language with existing pre-trained Language Models, as well as other types of bias

c) Well-written paper

**Reasons To Reject:**

a) Somewhat limited scope:
Even though the method is general enough to be applied to any kind of bias, it is only evaluated on nationality bias. This might be a bit limited for a full paper.

b) Somewhat anecdotal evaluation:
The paper shows many examples where the detected biases are consistent with results from political sciences - enough to convince me that the results are valid. However, for a fully reliable evaluation, it might be better to select one or two (source) languages or (target) nationalities and have all of the detected, associated biases evaluated by a domain expert (or, if possible multiple experts). I don't see this as absolutely necessary, but as something that would make the results even more convincing.

**Reproducibility:**

4: Could mostly reproduce the results, but there may be some variation because of sample variance or minor variations in their interpretation of the protocol or method.

**Reviewer Confidence:**

4: Quite sure. I tried to check the important points carefully. It's unlikely, though conceivable, that I missed something that should affect my ratings.

---

> ### Author Rebuttal · Authors · 2023-08-28
>
> We thank the reviewer for the valuable comments.
>
> > Using two-way sentiment classification instead of three-way classification:
>
> Two-way sentiment classifiers do not inherently introduce bias, as they have the
> capability to predict neutral inputs by giving approximately 0.5 probability
> scores for both positive and negative classes. We further analyzed this on
> 10,000 corpus examples in the English data with our sentiment analysis model. Out of these, 3,078 examples have
> neutral scores, falling within the range of 0.4<P(y=positive|x)<0.6. Therefore,
> we can say that our two-way sentiment classifier is able to make neutral
> predictions. We made an intentional decision in favor of two-way classifiers,
> since they would have better performance in detecting positive and negative ends
> of the sentiment spectrum
> (which is more important for our work) than three-way classifiers while
> understanding neutral instances as well.

---

### Meta-Review · Area_Chair_S7zU · 2023-09-22

**Recommendation:** 4

**Metareview:**

The authors proposed a method to detect the nationality bias present in a pre-trained Language Model. They fine-tuned LM on a sentiment corpus that is designed based on specific templates. Finally, they showed the bias in LMs while testing on a corpus containing nationality information.

Overall, reviewers agree that the work is sound, and contains interesting findings on bias in language models, though the application scope is limited (Reviewer qjcz and TUSA).

---

### Decision · Program_Chairs · 2023-10-07

**Decision:**

Accept-Findings

**Comment:**

The authors proposed a method to detect the nationality bias present in a pre-trained Language Model. They fine-tuned LM on a sentiment corpus that is designed based on specific templates. Finally, they showed the bias in LMs while testing on a corpus containing nationality information.

Overall, reviewers agree that the work is sound, and contains interesting findings on bias in language models, though the application scope is limited (Reviewer qjcz and TUSA).